# Opportunities to Strengthen Fish Supply Chain Policy to Improve External Food Environments for Nutrition in the Solomon Islands

**DOI:** 10.3390/foods12040900

**Published:** 2023-02-20

**Authors:** Senoveva Mauli, Anne-Marie Thow, Georgina Mulcahy, Grace Andrew, Anouk Ride, Jillian Tutuo

**Affiliations:** 1Australian National Centre for Ocean Resources and Security (ANCORS), University of Wollongong, Wollongong, NSW 2500, Australia; 2Menzies Centre for Health Policy and Economics, Charles Perkins Centre D17, University of Sydney, Sydney, NSW 2006, Australia; 3WorldFish, Honiara P.O. Box 438, Solomon Islands

**Keywords:** fisheries policies, nutrition sensitive fisheries, healthy diets, consumption, sustainability

## Abstract

Malnutrition and food insecurity have significant social and economic impacts in small island developing states, such as the Solomon Islands. Enhancing the domestic supply of fish, the main source of local protein, can contribute to improved nutrition and food security. This research aimed to improve understanding of the policy interface between the fisheries and health sectors and identify opportunities to strengthen fish supply chain policy to improve domestic (particularly urban) access to fish in the Solomon Islands. The study design drew on theories of policy learning and policy change and analysed policies using a consumption-oriented supply chain approach. Interviews were conducted with 12 key informants in the Solomon Islands, and 15 policy documents were analysed. Analysis of policy documents and interview data indicated that there were strengths as well as opportunities in the existing policy context. In particular, community-based fisheries management approaches and explicit recognition of the links between fisheries and nutrition were key strengths. Challenges included gaps in implementation, variations in capacities across government actors and communities, and limited attention to domestic monitoring and enforcement. Improving the effectiveness of resource management efforts may result in sustainable outcomes for both livelihoods and health, which will accomplish priorities at the national and sub-national levels and support the achievement of the Solomon Islands’ commitments to the Sustainable Development Goals.

## 1. Background

Malnutrition is a global public health issue [1,2] with substantial social and economic impacts, particularly on small islands developing states. In the Solomon Islands, malnutrition was the third leading risk factor for death and disability combined in 2017 [3]. The prevalence of undernourishment, a measure of food security, is at 11% [4]. 16% of Solomon children under the age of 5 years are underweight [5], and 32% are stunted in growth (<−2 Standard Deviation), with 10% being severely stunted (<3 Standard Deviation) [6]. According to the Demographic and Health Survey (DHS) conducted in 2015, micronutrient deficiency, particularly iron deficiency anaemia, is the third leading cause of disability in the Solomon Islands [4,6]. It has affected 40% of children under five years of age and 38–48% of women of reproductive age, increasing to 54% among pregnant women [4,6]. Non-communicable diseases (NCD) contribute to nearly 70% of all deaths, with poor diets as a major risk factor [7,8].

Studies show that the consumption of fish is associated with improved nutrition and NCD prevention [9,10,11]. The term ‘fish’ used in this paper refers to “seafood that includes fish or shellfish, divided into crustaceans and molluscs”. Crustaceans include shrimps, lobsters, crabs, and crayfish; molluscs include scallops, oysters, clams, and squid [11]. Fish are rich in multiple micronutrients such as vitamins A and E, minerals iron and zinc, and essential fatty acids and protein. The inclusion of fish in the diet contributes to dietary diversity and improved nutrition [12]. Policy to support increased production and local distribution of fish can, thus, play an important role in supporting improved food security and nutrition. In particular, adopting community-based fisheries management (CBFM) can support increased local access to fish [13].

The Solomon Islands is a stretched archipelago in the South Pacific that gained its Independence from the British in 1978. It is made up of about 1000 islands and six major islands, Choiseul, New Georgia, Guadalcanal, Malaita and Makira. The country has adopted a democratic system of governance which is called the “West Minister Model”. This three-fold model comprises a Parliament as the Legislature that is elected every four years, the Government as the Executive and the Judiciary. The Solomon Islands Constitution (“The Constitution”) is the supreme law of the Solomon Islands, and any other laws implemented in the country must be consistent with this Constitution. It is recognized in the Solomon Islands Constitution 1978 that acknowledges more than 90% of land and marine resources are managed under the customary marine tenure system (CMT). CMT is based on traditional marine knowledge [14,15] and is the traditional law [16].

Solomon Islands Government has recognised community-based approaches as the principal strategy in marine conservation and small-scale fishing management [17,18]. CBFM is recognised at all levels of governance (community, provincial and national) and to date, efforts across the Solomon Islands have involved communities in almost every province [19]. The Solomon Islands Government has also developed the National Ocean Policy, an integrated approach that combines all stakeholders in one policy space to discuss the management of oceans [20], and oceans are a policy focus for sustainability as described in the National Development Strategy 2016–2035.

However, despite strong policies for the management of fisheries and coastal marine resources broadly, the prevalence of high levels of malnutrition in the Solomon Islands indicates that more could be done to strengthen local access to fish, particularly in urban areas. Concerns are growing regarding shortfalls in fish production, with implications for achieving food security and nutrition objectives [21,22]. In addition, low dietary quality and diversity remain a challenge in the Solomon Islands, and fish and other seafood traditionally make an important contribution to nutrition [12]. Historically, and to date, there has been a focus by the national Government on fish exports and on fish as a livelihood strategy or ‘cash crop’ even at the local community level [23]. There is, thus, a need for further policy attention to the domestic supply of fish and the policies that govern the domestic supply of fish across sectors. Despite this focus on producing fish for the domestic supply, all the relevant government agencies, such as the Ministry of Environment, Climate Change and Disaster Management, Ministry of Development Planning and Aid Coordination, Ministry of Infrastructure Development and the Ministry of Commerce, Industries, Labour and Immigration have not been involved in the governing the domestic supply of fish in the region hence it has not been examined from a consumer perspective.

Strengthening the domestic supply of fish to combat all forms of malnutrition has been identified as a priority globally [24,25]. The aim of this study was to identify opportunities to strengthen policy in the Solomon Islands, such as that fish availability, price and marketing are improved for domestic consumers in urban areas. The objectives of this study were to improve the local fish supply chain to improve domestic access and affordability and improve nutrition through improvements to the food environment [26]. Our focus was on domestic supply chains rather than supply chains for export due to our focus on nutrition in the Solomon Islands.

## 2. Methods

We conducted a policy analysis, including documentary analysis and qualitative interviews with key stakeholders. To structure our policy analysis, we drew on a consumer-oriented supply chain analysis approach [27], which provides a focus on the consumer outcomes of supply chains. We drew on theories of policy learning and policy change to inform the study, particularly the development of instruments and analysis. These theories posit that policy change is influenced by the ideas and beliefs held by policymakers and other actors about the issue, the nature of the policy problem, and the existing policy and political context [28,29].

We identified existing policies relevant to fish supply chains in the Solomon Islands through internet surveys conducted from May to June 2019 on the nutrition and health, fisheries, environment, agriculture and development sector and direct requests to government officials. Documentary data were extracted into a matrix developed for the ACIAR project FIS-2016-300. The matrix comprises columns that list the core elements of fisheries management and governance that are stipulated in the sectoral policies within the Solomon Islands Government machinery [30]. We analysed policy content with reference to the policy analysis framework to identify policy strengths as well as potential opportunities for policy change to improve access to affordable fish.

Twelve key informants (9 males, 3 females) were interviewed from June to July 2019, representing government and donors, two from conservation non-government organizations (NGOs), two in related research institutions and one in industry and one each from the seven Government Ministries to name; Ministry of Environment, Climate Change and Disaster Management, Ministry of Fisheries and Marine Resources, Ministry of Development Planning and Aid Coordination, Ministry of Agriculture and Livestock, Ministry of Commerce, Industries, Labour and Immigration. Interviewers followed a semi-structured guide and asked key informants their views on policy challenges and successes, and opportunities for further action in relation to fish supply chains in the Solomon Islands.

### Analysis

Interviews were coded using qualitative software (NVivo^TM^) to check for commonly cited issues affecting all issues in the fish supply chain, including management, harvest, transport, storage, consumption and export. We first used the interview and documentary data to develop an overview of policy relevant to the ‘domestic and export-oriented consumption supply chain’, identifying for each point of the supply chain the core activities, key stakeholders and relevant policy documents. The analysis of coded interview data focused on six areas (see Table 1 below).

## 3. Results

### 3.1. Overview

The analysis of fish supply chain policy identified a number of strengths of the fish-related policy landscape in the Solomon Islands, including a consistent policy focus on community-based resource management and explicit policy recognition of the links between fish and nutrition. However, there were points of incoherence across sectors and national fish supply policy goals, governance, and health-focused perspectives on nutrition.

### 3.2. Current Policy Context

The National Development Strategy (NDS) is an overarching policy document that includes fisheries within a broader national development framework. It articulates the Government’s commitment to alleviating poverty across the whole of the Solomon Islands, to equity, and to ensuring basic needs are addressed, and food security is improved (Table 1). The NDS specifically identifies fisheries as a strategic sector for development due to its contribution to economic growth and food security.

The Solomon Islands Fisheries Policy 2019–2029 guides the production, conservation, management, development and sustainable use of fisheries and aquatic resources of the Solomon Islands (Table 1). The Ministry of Fisheries and Marine Resources (MFMR) is mandated with responsibilities to manage and develop fisheries and aquatic resources across the Solomon Islands’ including coastal areas, where local indigenous peoples practice community-based fisheries management with the support of the government. Other areas of responsibility also span inland freshwater fisheries and aquaculture and across the Exclusive Economic Zone, which is a source of tuna for canning and local consumption and purse seine and longline caught fish for export. Management for these resources comes under the Fisheries Management Act 2015. Other documents key informants reported as relevant to fisheries included the Regional Action Plan and National Action Plan of the Coral Triangle Initiative, Inshore Fisheries Regulations, the Solomon Islands Fisheries and Marine Reserves Strategy, and the National Oceans Policy.

The conservation and management measures of the Western and Central Pacific Fisheries Commission, a regional institution to manage tuna and other billfish across the region, were also identified as important policy support. The environmental policy also supports sustainable fisheries in the Solomon Islands, namely The National Biodiversity Strategic Action Plan 2016–2020, and the National Trade Policy Framework explicitly supports these objectives by prioritising sustainability within the fisheries sector.

At the national policy level, transport, infrastructure and industry policies also play a role in supporting domestic access to fish (Table 1). Priorities for transport infrastructure, small businesses and access to markets contribute to livelihoods as well as increased access to fish in urban areas. These policies are complemented by municipal regulation of formal markets, while informal roadside and village markets are very rarely regulated by authorities.

The Lokol Kaikai Initiative includes a priority for ensuring the continuous contribution of fisheries to the local diet, conservation of stocks and promoting sustainable management of fisheries resources (Table 2). The agriculture sector is responsible for local food consumption and food security needs and acts as lead for the ‘Lokol Kaikai initiative’, whilst MFMR takes the lead in fisheries production and consumption. The Ministry of Health and Medical Services and the Ministry of Education and Human Resources Development are also lead institutions for this initiative, with responsibility for nutrition-related service provision as well as education and promotion. These policies and initiatives indicate a national policy focus on the fish supply chain- food production, management, and export (for commercial). They acknowledge the interconnected nature of fisheries, local fish supplies and growing rates of non-communicable diseases as fresh fish prices increase. A key strength of the Lokol Kaikai Initiative is its explicit articulation of the link between nutrition policy and fisheries. Similarly, SIG has specifically identified the relationship between fish production, nutrition and food security in the National Food Security, Food Safety and Nutrition Policy 2019–2023 (*draft*). Policy priorities include safeguarding biodiversity and improving the quality and quantity of local fish production, climate change, and diversification of local food production to ensure the availability, accessibility, and affordability of safe and nutritious foods to Solomon Islanders.

### 3.3. Successes

In terms of successful interventions to improve fish supply, the most commonly cited success in the interviews was the implementation of Community Based Fisheries Management (CBFM) efforts under which local villages managed their coastal areas. 67% of interviewees (8 participants) talked about CBFM when asked about successes in fish supply. The second most noted change was increased capacity in the Provinces. The Provincial Governments are established under the Provincial Government Act 1989, and they are mandated as agents of the National Government. They have devolving functions for the fisheries sector, so a degree-holding staff is recruited to manage the Provincial Fisheries office. Regular meetings are held by the Provincial Fisheries Officer to coordinate activities leading to a stronger enabling environment across all three levels of governance (community, provincial and national). One government informant outlined the major changes:


*With the new recruitment, we have been able to have one officer for each province to implement Community-Based Resource Management (CBRM). This new strategy of having an officer in every province, we have seen that there is progress…Before there wasn’t any proper guide to facilitate training and awareness, now we have the facilitators guide, we have been able to facilitate in a well-organised manner to communities.*
(Government informant)

External partners such as the national Government and international or locally based NGOs often provide support in the form of information, assisting communities in setting up and monitoring their management regimes. Strengthening of the regulatory environment to enable CBFM was reported also to support improved management at the community level; for example:


*In the fisheries act, … a major change that impacts my division is the inclusion of community empowerment to manage their own resources—I think having that legal framework available is a platform for communities to really come up with their own bylaws in terms of managing their resources.*
(Government interviewee)

Seven (7) informants commented that the capacity and will of the communities to manage their resources had improved. In the past years, attention has been on resource management efforts, but the communities implementing these activities also questioned the livelihood benefits of this work. In recent years, some NGOs supported communities with activities such as Savings Clubs. Saving Clubs are created in the community as microfinancing entities where community members (mostly women) who are registered as its members meet at an agreed time to allocate funds they intend to save from their daily earnings from their resources. Community members see these activities as tangible economic benefits of doing resource management:


*We have been trying to get them to manage it (our resources), but because they rely upon it (the resources) so much, they have been overharvesting, but now that they have the microfinancing component (the Saving Club). Now if you go there, they have closures and bag limits of how much you can harvest because they know now from the money it has generated and the savings that the women have made. Now they understand that you have to manage it properly to make the money they are making now. So that is a good example.*
(NGO interviewee)


*Resources are owned by the people, and that is enshrined in the Solomon Islands constitution; we all recognise it, and that is also how people perceive these resources that are there... I think that has been done quite well in terms of the state playing the role that it can play and the communities playing the role that they can. I think there has been significant advances recently in that engagement.*
(Government interviewee)

At least four sites have undergone the required legal process described in the Protected Area Act 2020, which also includes CBFM rules. These sites have been gazetted and are recognised under the status of law [31,32]. There are a few Provinces that have local regulations and procedures in place through Provincial ordinances and are encouraging communities to continue their local management regimes [31]. Despite limited government capacity, communities were taking action to implement CBFM, and enforcement of CBFM at the community level was largely reliant on the communities’ own efforts, for example:


*…here is always this question about government regulation policies vs. traditional resource ownership and customary rights and so forth. So, I know a lot of communities still hold onto those traditional rights (Customary Marine Tenure rights) very strongly, they in some ways, you see a lot of communities that have gone ahead and done their own management because they have waited so long for something to happen and nothing has eventuated. You will find that those communities still use those traditional managements and their own rules and regulations, and they probably are the best ones, out of all the other communities that have established these small managed areas waiting for policy and stuff to back it up.*
(NGO interviewee)

Two government interviewees mentioned aquaculture as an achievement, particularly the setting up of an aquaculture division and building of a hatchery; previous efforts were around the Mozambique tilapia in the highlands. The Mozambique tilapia was introduced in the 1950s and 1960s; however, it has been considered to perform poorly in aquaculture. The current focus has been on Nile Tilapia or milkfish because it is widely introduced in the Pacific, and the Solomon Islands Government sees its potential for the future supply of the domestic fish markets. However, aquaculture is still on a very small scale and has not yet made an impact on aquatic food supply chains.

### 3.4. Policy Challenges

Most key informants pointed to gaps in implementation and sometimes the political will to carry through policy objectives into action. There were 22 references across the interviews to lack of regulation or enforcement as being an obstacle to optimal management and use of fish as seafood. One interviewee, in describing formal supply chains (for tuna for export or canning and local sale) and informal supply chains (coolers sent from one part of the islands to another for sale and local consumption), commented, *“none of the supply chains are particularly regulated at all”* (Government interviewee). Another NGO interviewee identified a lack of enforcement of laws around species under threat.


*National regulations to harvest and sell turtles (which are prohibited species stipulated in the Fisheries Act 2015), people are not allowed to harvest nests, but our finding indicates that they continue to do those things despite the regulations in place. I think it is something to do about implementing reinforcement of the regulations. We have good policies there, but the lack of enforcement and implementation is the issue.*
(NGO interviewee)

Two interviewees identified the lack of enforcement of standards for the size and quality of fish in local markets, which in turn posed health risks for consumers of damaged fish. A government interviewee identified the practice of “salt fish” (purse seine fish stored in brine that was unfit for export because of their state they can be regarded as a food safety risk and, therefore, sold or bartered to islanders for sale at local food markets) to illustrate the lack of enforced standards. Rule breaking regarding fisheries was mostly caused by people’s needs or desires for harvests to make money, interviewees said, as well as, in some cases, sabotage from neighbouring communities not involved in making the rules. One NGO interviewee said the mismatch between the aims and approaches of conservation organisations and the livelihood and nutritional needs of communities had been an obstacle to progress in fisheries management in the past. In addition, the lack of enforcement of standards was related to a lack of licencing; for example:


*The eskies are individually operated and out of our scope. … But this is where we want to work with the provincial government for them to have more say in that in terms of licensing people who are operating in the province. …*
(Government interviewee)

Interviewees contrasted the well-developed policy and monitoring frameworks for export with the less-developed frameworks for domestic supply. For the export of tuna, measures such as fisheries observers on boats, designated ports for landing, cameras and electronic monitoring are in place due to regional efforts to control the tuna resource. In contrast, local catches of pelagic invertebrates and coastal fish could be landed anywhere with relatively little monitoring or information collection in place. An interviewee from academia attributed the perceived lack of regulation tools, approaches and enforcement to a priority for the economic interests related to offshore fishing. They observed that the offshore stakeholders had greater capacity and resources than those in coastal fisheries, with the latter operating with limited knowledge and control of the supply chain. Interviewees identified the lack of specific policies regarding domestic marketing for fish as a challenge for domestic supply; for example:


*How do we help local communities to market their harvest or their catch?... we don’t have any clear and any well-developed market structure at the moment; it is not there. Let me correct myself, for fisheries. Generally, it is there, but for fisheries, it is not there. Poor infrastructure, I am talking about post-harvest infrastructure. Good access to transport and communication, networks and all that kind of stuff. We need these in place.*
(Academic interviewee)

The issue of mining and logging affecting marine resources was also raised by NGOs as an issue of concern, with the lack of political will to set and enforce environmental standards and lack of regulation of the sector being cited as issues holding up progress:


*Marine Protected Areas are great tools, but for some reason, I don’t think they have really worked well in the Pacific, and one of the reasons is Melanesian culture and resource ownership and traditional ownership, and they really depend on these areas… NGOs are trying to, I guess to push this MPA approaches. So rules are broken, not because… they don’t agree with the concepts of MPAs, but their dependency on the fisheries around these areas is so high that I guess they don’t stick within the rules of these managed areas.*
(NGO interviewee)

MFMR, as the national agency mandated for fisheries management and development, also faces challenges in regulating an archipelago of islands because of limited staff and resources. Their enforcement capacity was perceived as further reduced by the limited funds for coastal fisheries. For example, one NGO interviewee commented:


*I would say 90% of the fish down at the market would be illegal—size related. Even how these fish are being caught—undersized nets, and the certain harvesting of particular species. For example, the Napoleon wrasse are an endangered/protected species, but those things aren’t being enforced. But also, it is very hard here in the Solomon’s in the sense that, because it is so spread out, and it is a huge challenge for the government to really focus on these policies. I think they are a little too centralised as well; a lot of these policies and regulations should be more pushed out to the provinces.*


The failure of policy and implementation to deliver local benefits was the second most common challenge for the Solomon Islands and fish supply chains that were identified by interviewees. National government policymaking was seen as removed from the realities of coastal communities; elected leaders and officials were involved but not, for example, chiefs and women leaders. Related to this was concerns about who was driving development, foreign interests with a focus on offshore fisheries or indigenous resource owners and businesses; for example:


*I do not think fisheries supply chains are a priority because it is not seen as in the forefront of livelihoods activities. Compared to land-based activities. For example, inshore fisheries are mostly seen as a subsistence activity rather than promoted as a business.*
(Industry interviewee)


*It is not enough to say we need indigenous participation. For instance, what does it mean? Does it mean you are going to work on post-harvest challenges? It doesn’t spell out that. Does it mean you are going to work with indigenous communities to find ways of diversifying markets, identifying markets and diversifying them to be able to take advantage of?*
(Academic interviewee)

The urban supply of fish is also constrained by infrastructure-related challenges. The distance of harvest sites from the main population centres is a key constraint. Exporting produce to Honiara, which could be an opportunity to increase sales and income, is too expensive, owing both to the expense of exporting and a lack of cost-effective and regular transport. Another key NGO informant said starting up exports of fish suffered the same constraints:


*…there used to be a fish buyer here… I think one of the things he had challenges with is the expense and back then, he didn’t have the transportation, there weren’t regular flights out, and there wasn’t any infrastructure in place, but even for a business to set something up like that, any business here is very expensive to start up, for example, electricity costs are very expensive, i.e., to run snap freezers, the cost would be huge. And then the other problem he had was he just couldn’t meet the demand from what the suppliers were wanting.*
(NGO interviewee)

Despite the policy priority for domestic value adding, fish supply chains were described as highly localised and poorly resourced. As one government interviewee explained:


*You will have fishers who have to cook to fish so that they can be preserved so that they can travel a bit further, for example, across Gela, in some places, there is abundant fish, and they do not have ice…so what they do is they cook the fish, either by baking in the stone or barbecuing them and racing them straight to [where people pay], so that is a part of the value chain. [and] you have women who reside in the urban areas buying fish, and then cooking them and either selling them again at the market or providing them as part of the catering when they are catering a meeting.*
(Government interviewee)

Interviewees mentioned that for islands in the furthest North East of the country, such as Choiseul and Shortland Islands, it would be easier to sell fish to Bougainville (on the PNG border to the West). Similarly, in islands in the farthest East, such as the Temotu Province, it would be easier to sell fish to Vanuatu. Half of the interviewees identified preservation and transport challenges. These included a lack of facilities and skills to preserve fish (and, therefore, transport it to markets further afield).


*Most of the fish caught, or most of the sea produce harvested, are consumed instantly, and that is not very good enough in the sense that in the case, for some reason, a week or two, you are not able to go out into the sea to harvest, what do you eat?*
(Academic interviewee)

Interviewees described government-established fisheries centres as sites where people could preserve, sell and ship fish as a key intervention. However, several interviewees noted that the politicisation of the sites and lack of consideration of market needs meant the centres were not functioning as intended in many sites; for example:


*I think what is happening is that we continue to suffer two things: first is the scale of the economy—not everyone will want to go fishing every day, so for ice, the generators will need x amount of petrol every day, so whether you have one cube of ice in there or you have several blocks of ice … The second issue is the tyranny of distance and that automatically adds on to expenses because you need petrol to move fish from place A to place B …So, I see those as two of the main issues which affect the success of fisheries centres and the policy which is trying to facilitate value chains, supply and distribution of fish. There are some places where no matter how many fisheries centres you build; they will not succeed, boats don’t go there all the time.*
(Government interviewee)

On the consumer access side, a key policy concern raised by half of the interviewees was the scarcity of fish. However, in a way, this scarcity was hidden from view as it was localised and complicated by geographical challenges. The Solomon Islands have archipelagic waters with urban settlements spread wide amongst the island chain creating significant distances to transport produce across. For example, an NGO interviewee stated: *“[consumption of fish] depends on the geography. 75% of the meal will be fish in Choiseul compared to once per week in Guadalcanal”.* Another government informant said the scarcity or supply of fish was dependent on distance from a market from sales (remote communities having more fish, as fewer opportunities to sell fish), local population compared to aquatic resources, and religious and cultural limitations which influence which species people eat or not eat:


*I think there is enough fish; the problem we have is distribution. So, although there might be less amount in one part, there is always more in another, so you have to move the fish from one place to another, so looking at the whole thing in totality, and just the population and the amount of fish, there is sufficient fish for everyone per capita, the issue is distribution and providing fish where it is needed.*
(Government interviewee)

Concerns about scarcity were also identified by industry interviewees, who commented that nutrition studies indicated that people did not have enough protein in their diets and that sales of canned tuna indicated that people were not eating fish as regularly as required. In addition, some NGO interviewees noted stunting and malnutrition in some rural and remote communities and practices such as selling fish for money rather than eating them for good health; all these factors are potentially related to poor nutrition. One government interviewee noticed a change over their lifetime where before, fish was easily accessible close to the shore and cheaper than chicken; now, chicken was more affordable than fish in Honiara, and fishers have to go further and for longer to catch fish.

### 3.5. Opportunities to Strengthen Policy to Improve Access to Affordable Fish in the Solomon Islands

Most interviewees identified opportunities to improve the domestic supply of fish through scaling up CBFM, improving enforcement, utilising management for economic opportunities, and government (national and provincial) being better linked with communities to provide the right advice and support. Several interviewees also mentioned that better enforcement and monitoring of offshore fisheries could also result in local benefits, as many species such as turtles, sharks and other fish apart from tuna were being caught, and this impacts the local food supply.

Overall, there were three themes that emerged about how these outcomes could be achieved for fish supply chains in the Solomon Islands.

First, enforcement needs to be addressed through the implementation of CBFM approaches. There are mounting interests from communities, so there is potential to encourage participation in CBRM approaches further so they can sustainably manage and, as a result, even increase the supply of aquatic foods. Building on the significant policy interest in communities sustainably managing and developing their aquatic foods to increase the local supply and distribution of fish, there is an opportunity for a renewed focus on the benefits of communities being nourished by their local environment—including urban communities.

Second, there is an opportunity to align fisheries-related policies at all levels (community, provincial and national) and for collaboration in policy implementation.


*I think it’s not so much about policy, people in the community don’t worry about policy; it’s more for awareness and the real implementation and how best we can do that. We are not really working on our part; we need to strengthen our relationship with the fishers and the fishing associations.*
(Government interviewee)

Interviewees observed that relevant government agencies at the national level are working in silos and that links to provincial and community implementation activities and projects were sector based and rarely integrated. By addressing these issues of governance and building those relationships, there is potential to achieve cross-sectoral objectives, including livelihoods, environment and nutrition.

Third, interviewees identified an opportunity for increased investment in post-harvest processing and strategically located infrastructure to address the supply chain challenge described above. Building on existing government initiatives, such as fisheries centres, to increase access to training and processing facilities could increase access to the growing urban market. Prioritising policies for the preservation and transport of fish could be the most useful intervention to increase food supply and distribute it more evenly across regions within the Solomon Islands, considering that the transport costs and distances across the islands were seen as beyond anyone’s influence.


*So there should be emphasis or effort to try and ensure that at least fishers are taught how to preserve their harvests either for their own personal use, for home use, or for the market. Because there are markets, particularly the Asian market, that enjoy preserved or even smoked fish, for instance. Here I can hardly find smoked fish.*
(Academic interviewee)

## 4. Discussion

Fish represents the major protein source for local populations in many regions of the world [33]. This is certainly the case for the Solomon Islands, where more than 80% of its local population live in rural areas [13,34,35]. The Solomon Islands is a small island developing state in which fish makes a significant contribution to the economy as well as food security and nutrition [23]. This study has provided insights for policy in other SIDS and countries with significant domestic fisheries industries, in which malnutrition remains a challenge. In Eastern Africa, for example, domestic fish supply is hindered by post-harvest losses, poor infrastructure and market linkages [36].

The Solomon Islands’ local communities are heavily dependent on marine resources for their food, which is a means of livelihood security [37]. However, access to local fish in the urban areas (for instance, the Provincial Centres and Honiara) is increasingly challenging. In line with global recognition of the importance of food systems for nutrition—and the potential to transform local food systems in ways that balance economic, environmental and nutritional imperatives—this paper examines local fish supply as a policy challenge for the Solomon Islands.

This analysis of fish supply chain policy identified three key strengths of the fish-related policy landscape in the Solomon Islands. First, the consistent policy focus on CBFM was uniformly identified as a positive by the interviewees. This aligns with other research showing that community-based resource management is contributing to better environmental and livelihood outcomes [13,38,39,40]. We also found that policy implementation was supported by external partners such as NGOs, which has been similarly identified by other studies as strengthening implementation [41,42,43]. The relevant Government agencies may not be well-resourced, so they would encounter difficulties in accessing remote communities across the country. NGOs are contributing to implementing activities at the community level, particularly on CBFM efforts. Examples include the Nature Conservancy, which is implementing projects in Choiseul [44] and Isabel Province [44] with an emphasis on practising conservation to provide evidence-based results such as stock increase. The Worldwide Fund for Nature (WWF) [45] is implementing CBFM activities in part of the Western Province, and their approach is addressing sustainability through microfinance and practices of CBFM. Such collaborations seem to be resulting in really good CBFM practices over the last two decades.

Second, the explicit policy recognition of the links between fish and nutrition created a basis for collaboration between the health and fisheries sectors. This integration reflects global best practices for both fisheries and nutrition policy and provides a clear example of how recommendations for cross-sectoral engagement can be translated into specific sectoral policy at the national level. For example, the operationalisation of recommendations within the United Nations Voluntary Guidelines on Food Systems and Nutrition for Sustainable food supply chains to achieve healthy diets [46].

Third, the Ministry of Fisheries and Marine Resources, in collaboration with the Ministry of Environment, Climate Change, Disaster Management and Meteorology, has established policy coordination mechanisms for other sectors that might be relevant to nutrition. National platforms in the environment and fisheries, such as the National Coordinating Committee, and integrated policy frameworks, such as the National Ocean Policy 2018. The National Coordinating Committee was established as a requirement for all member countries of the Coral Triangle Initiative (which includes the Solomon Islands). National agencies are required to integrate into discussions on the marine space; hence, there is the opportunity to have joint discussions on policy implementations. A recent policy document that can also bring the relevant agencies together is the National Ocean Policy. This integrated policy framework provides an avenue for agencies to interact and an opportunity to align policies together.

Despite these strengths, the study also found—similar to findings from other Pacific Islands countries and in other regions—that food system policies are not always aligned to incentivise improved nutrition [47,48]. Although community-based efforts are widespread in many parts of the country, the ongoing challenge is enforcement at the community level. Community-based studies have also found that management rules are not always followed, and because of family relationships in the same community, there is a challenge when imposing penalties [49,50]. Similarly, enforcement challenges at the community level have also been observed in studies of environmental regulation [51]. Conserved or managed areas may be situated far from the community, for example, beyond the foreshore reefs or up in the highlands, so accessing these areas and imposing full-time management measures can be costly to the people [52].

Overall, this study suggests that stronger and more coordinated policy implementation is needed to address the pervasive challenges in supply chains that are hampering access to fish, particularly for urban consumers. Over the years, national-level initiatives have involved government projects through relevant agencies such as MFMR, including the establishment of Provincial Fisheries Centres at identified locations across the country. However, there is mounting evidence that these projects have not succeeded over the years [31]. Those managing the centres may lack the technical knowledge to maintain the facility, there may be ongoing disputes regarding the ownership and management of these facilities, and, most importantly, connecting these facilities to market outlets in the Provincial Centres and to Honiara has failed [53,54]. Elsewhere in the Pacific, similar challenges to successful and effective implementation have also been identified [55]. In Fiji, for example, addressing implementation was the leading issue in public sector reforms [56]. There may also be opportunities to collaborate with the private sector, which may be able to contribute because they have the proper facilities to preserve fish in remote parts of the country and would also have more reliable transportation to market centres [57].

We also found a pervasive emphasis on economic outcomes at the policy and community level, which may come at the expense of improving nutrition outcomes in communities. Other research has found that people prioritise the economic benefits and are motivated to so they sell their best harvest, leaving the smaller-sized fish at home for consumption [12]. The limited access to urban areas may also be from food preference, which results in low nutrition intake in the community. It is common practice that people sell their harvest and, in turn, purchase processed tinned fish, noodles and rice at the community canteens or the shops in the Provincial Centres. Locals may prefer canned fish because it is more convenient, cheaper, has a long shelf life and is less perishable [58,59,60]. In general, communities across the Solomon Islands are experiencing dietary transitions [60] as the result of processed rather than fresh foods [61].

## 5. Study Limitations & Strengths

The main limitation of this study is the limited number of interviews conducted. The stakeholders who participated in the interviews likely influenced the themes that were identified. However, the approach to interviewee selection was to be representative of decision-makers in different sectors, government, industry, and others rather than representative of populations as a whole. Integrated analysis and review of policy content likely provided strength to the study.

## 6. Conclusions

Strengthening food systems to improve nutrition is a global priority. In this study, by analysing fish policy and implementation, we were able to identify opportunities to enhance the domestic supply of fish, a major source of dietary protein to contribute to improved nutrition. Key strengths in the policy context included community-based fisheries management approaches and explicit recognition of the links between fisheries and nutrition. Challenges included gaps in implementation, variations in capacities across government actors and communities, and limited attention to domestic monitoring and enforcement. Expanding community-based fisheries management, as well as policy coordination and implementation, are key opportunities to improve sustainable outcomes for livelihoods, environments and nutrition.

## Figures and Tables

**Table 1 foods-12-00900-t001:** Codes used to analyze the interview data.

Codes	Description
Supply chain features	Community-based fishing, storage/distribution, processing, retail (locally oriented and export-oriented)
Frames/beliefs—local fish supply	Nutrition, sufficient supply, importance of fish, fish development an issue
Governance structures	For fish supply, sectoral responsibilities (for fish-related policy)
Government policies	Objectives and activities related to fish supply
Policy implementation	Successes and challenges
Opportunities	To strengthen policy to improve access to affordable fish

**Table 2 foods-12-00900-t002:** Key policy sectors and policy documents relevant to fish.

Relevant Sectors	Title of Policy	Stated Objectives of the Policy	Year Endorsed	Ministry Responsible
**Whole-of-Government**	National Development Strategy 2016–2035	- “Sustained and inclusive economic growth” - “Poverty alleviated across the whole of the Solomon Islands, basic needs were addressed, and food security improved; benefits of development more equitably distributed” - “All Solomon Islanders have access to quality health and education” - “Resilient and environmentally sustainable development with effective disaster risk management, response and recovery” - “Unified nation with stable and effective governance and public order”	2016	Ministry of Development, Planning and Aid Coordination
**Whole-of-Government**	DCGA Policy Translation and Strategy	“Provides the Strategies and Intended Outputs or deliverables from which line ministries will develop their Corporate Plans and Annual Work Plans for implementation to achieve the intended Policy Outcomes in the DCGA Policy Statement.”	2019	Prime Minister’s Office
**Fisheries**	Fisheries Management Act 2015 & Regulations	- “Ensure the long-term management, conservation, development and sustainable use of Solomon Islands fisheries and marine ecosystems for the benefit of the people of Solomon Islands.” - “Fisheries resources are to be used sustainably so as to achieve socio-economic benefits, including economic growth, human resource development, employment creation and sound ecological balance, consistent with Solomon Islands’ national development objectives”	2015	Ministry of Fisheries and Marine Resources
**Fisheries**	Solomon Islands National Fisheries Policy 2019–2029	A policy for the conservation, management, development and sustainable use of the fisheries and aquatic resources of the Solomon Islands—socioeconomic needs	2019–2029	Ministry of Fisheries and Marine Resources
**Fisheries**	Solomon Islands National Aquaculture Management and Development Plan 2018–2023	Aquaculture can play an important role in social and economic development, as well as food security and the livelihood of the people. This Plan is a roadmap that sets out clear and comprehensive objectives that are supported by MFMR strategies.	2017–2019	Ministry of Fisheries and Marine Resources
**Environment**	The National Biodiversity Strategic Action Plan 2016–2020	[For fish]: protecting and managing marine and coastal biodiversity as well as the protection of indigenous knowledge	2016–2020	Ministry of Environment, Climate Change, Disaster Management and Meteorology
**Trade**	The Solomon Islands Trade Policy Framework and the Solomon Islands Trade Policy Statement	Build the productive capacity of the Solomon Islands economy via sustainable trade and investment. The resulting creation of wealth and employment opportunities is aimed at promoting human development, reducing poverty and improving living standards for Solomon Islanders. Includes priority for sustainable fisheries.		Department of External Trade/Department of Foreign Affairs and External Trade
**Transport, Infrastructure**	National Transport Plan	Facilitation of goods and services and interaction between communities—vision: effective transport infrastructure and transport services to support sustained economic growth and social development in the Solomon Islands	2010–2030	Ministry of Infrastructure and Development
**Transport, Infrastructure**	Solomon Islands National Infrastructure Investment Plan	Improving the infrastructure of the Solomon Islands—no specific Vision or Mission Statement included	2013–2023	Ministry of Development, Planning and Aid Coordination
**Industry**	MCILI Corporate Plan	Empowering Solomon Islanders, reserved business areas for Solomon Islanders, economic reforms, productive economic sector, infrastructure development, foreign relations, education	2016–2019	Ministry of Commerce, Industries, Labour and Immigration
**Industry**	Micro, Small and Medium Enterprises (SMEs) policy and strategy	Sectoral ministries should be responsible for market-based research and development, knowledge transfer, efficient and sustainable production tech		Ministry of Commerce, Industries, Labour and Immigration
**Marketing**	Honiara City Council (Markets) Bill 2009	“Regulation of public markets in Honiara City and for related purposes”	2010	Honiara City Council
**Nutrition**	National Food Security, Food Safety and Nutrition Policy 2019–2023 (draft)	Vision statement: achieving food and nutrition security in the Solomon Islands through long-term, sustainable collaboration and engagement by all key stakeholders Goal: to ensure sufficient, safe, nutritious foods are readily available, accessible, affordable and acceptable to all Solomon Islanders at all times	2019–2023	Ministry of Health and Medical Services, Ministry of Agriculture and Livestock Development, Ministry of Fisheries and Marine Resources, Ministry of Education and Human Resources Development
**Nutrition**	Lokol Kaikai Initiative	The Solomon Islands is a food and nutrition-secure country supported by a modern, resilient, evidenced-based and sustainable agriculture sector for the population.//diversified local food production that is accessible and affordable that contributes to a strategy that achieves food and nutrition security requirements for all Solomon Islanders	2019–2023	Ministry of Agriculture and Livestock

## Data Availability

Data is contained within the article.

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
