# Peer review of "Opportunities to Strengthen Fish Supply Chain Policy to Improve External Food Environments for Nutrition in the Solomon Islands"

_foods, 2023, doi:10.3390/foods12040900_

Round 1

Reviewer 1 Report

This manuscript is about the availability of fish and other aquatic organisms to local populations in the Solomon Islands.  The data include a detailed description of relevant formal policies (Table 1) and the results of interviews of 12 individuals (given as partial transcripts interspersed with narrative comments).  I have several general suggestions for improvement, one of which is adoption of a standard definition for fish (details below). 

Another general suggestion is to provide more background information on the Solomon Islands, a location that will not be familiar to most readers.  This would help, for example, in placing the discussion of provinces into proper context.  Context is also lacking in the treatment of an issue involving two kinds of tilapia, “Melanesian culture,” and the effects of mining and logging on fish.  The authors should try to conceptualize these and other issues through the eyes of an interested reader who has never visited the islands and knows nothing about their administrative organization and other issues/controversies that may be very familiar to those with local knowledge.

I also strongly recommend that the authors reorganize the results and add a table to condense the long narrative (six full pages), which includes lengthy transcripts and often contains extraneous information that is difficult to understand.  Six foci are identified in Section 2.4, but these are not followed as organizational principles in the Results Section.  These should be used as headings in a table, with shorter excerpts (distilled from the long excerpts) and comments added within the table—and more detailed information added outside of the table.  This approach would be vastly superior to the current approach, which forces the reader to plow through pages and pages of information and try to distill out the important points and understand them with respect to the six foci introduced in Section 2.4.    

Specific comments are as follows:

L16.  Malnutrition and food insecurity have..

L46.  Please explain the SD abbreviations used here.

Paragraph beginning at L54.  Your unusual and expanded definition of fish to include other aquatic animals creates a lot of ambiguity, if for no other reason than the fact that these other aquatic organisms are not fish (it is almost as if an author would state that pastures contain cows, sheep and goats but that for the purposes of a manuscript, they are all considered to be cows!).  Moreover, if you consider other aquatic organisms to be fish, you create significant difficulties in interpreting the literature.  For example, the first sentence in this paragraph cites three references.  Ref. 9 and 10 consider fish to be fish, and Ref. 11 considers fish to be one component of seafood.  How can you relate your work to these and other references if your definition of fish is non-standard?  I strongly recommend that you follow the example set in Ref. 11 and define all of these organisms as seafood.  It would be fine for you to point out that most of the organisms classified as seafood are fish.  Then in the subsequent text, you can use the word fish when you mean fish and seafood when you mean fish and other aquatic organisms. 

L66.  SIG is an unneeded abbreviation.  It is only used three times and should be deleted.  Just spell out the phrase.

L67.  Principal

L71.  Add a reference describing the National Ocean Policy.

L89/92.  Which government agencies do you mean?  Be more specific.

L99.  Most readers will know little or nothing about the Solomon Islands, and so it would be helpful to provide more information here, especially about provinces, a concept that is discussed later.

L99/102.  SIDS is another unnecessary abbreviation that is only used twice.  Just spell out the phrase.

Methods.  There is no need to subdivide this relatively short section into four subsections, each containing just one paragraph.  Consider removing all of the subsections and just providing the reader with a fluid, uninterrupted narrative.

Paragraph beginning at L107.  This belongs at the end of the Introduction—not in Methods. 

Paragraph beginning at L121.  You need much more detail here.  You must tell us exactly which internet searches were conducted, when they were conducted—and you must define the relevant sectors.  Provide at least some details about the matrix and the study framework, too. 

Paragraph beginning at L130.  More detail is needed here.  There were 12 informants, and you give us information about five of them.  What about the others?  What organizations do they represent?  Also, it was interviewers (not interviews) who followed and asked.

L143.  What is the rationale for subdividing the supply chain into domestic and export-oriented components? 

L147.  See above comment about your definition of fish.  Adding invertebrates here adds yet another element of confusion (your definition of fish includes invertebrates).

L148.  The reader does not know why you have bracketed the word processing.

L158/161.  This information belongs at the end of the manuscript.

L240 and L249.  CBFM has already been defined—no need to define it again here, just use the abbreviation.

L244/246.  Most readers will have no idea of the provincial structure of the Solomon Islands.  See earlier comment.

L252.  You have already stated at L246 that this individual is from the government.  The word interviewee is preferable to informant.

L264.  Readers understand what is meant by seven—no need to write 7, but what do you mean by “about?”  With such a small number of interviewees, you should be able to give us an exact number. 

L268.  What is a Savings Club?

Information beginning at L271.  What is “it” in the first sentence, and what is meant by “they have the microfinancing component to, now if you go there?”  You can add parenthetical information to help make ambiguities such as these understandable to the reader.

L292.  What is meant by those rights?  Which rights?

L300/301.  The reader doesn’t understand the issue about Mozambique vs. Nile tilapia.

L308.  See earlier comments about definitions.

L310.  What is an eskie?  The internet has several definitions, none of which seems to fit with this manuscript.

L315.  Based on previous definitions and narrative, turtles are not within the scope of your study.

L554.  No need to re-introduce this abbreviation.

L558/559.  No need at all for the abbreviation NCC.

Reviewer 2 Report

The MS is very interesting and deserves interest given that the subject is relevant to the area of study. But the MS is a little confused in some parts (especially results) and a revision is required. To give an example, the results of the interviews can be easily moved to an appendix in order to not make the main text so long and confusing. Please find the detailed comments on the attached pdf 

Round 2

Reviewer 1 Report

The revised version is acceptable to me.